# Using TLC-MALDI-TOF to Interrogate In Vitro Peptidyl Proximal Preferences of PARP14 and Glycohydrolase Specificity

**DOI:** 10.3390/molecules28166061

**Published:** 2023-08-15

**Authors:** Zeeshan Javed, Hannah H. Nguyen, Kiana K. Harker, Christian M. Mohr, Pia Vano, Sean R. Wallace, Clarissa Silvers, Colin Sim, Soumya Turumella, Ally Flinn, Anthony Moritz, Ian Carter-O’Connell

**Affiliations:** Department of Chemistry and Biochemistry, Santa Clara University, Santa Clara, CA 95053, USAcmohr@scu.edu (C.M.M.); pvano@scu.edu (P.V.);

**Keywords:** PARP, PARP14, ADP-Ribose, ARTD, ARTD8, NAD^+^, TLC-MALDI-TOF, ADP–ribose glycohydrolase

## Abstract

The transfer of ADP–ribose (ADPr) from nicotinamide adenine dinucleotide (NAD^+^) to target proteins is mediated by a class of human diphtheria toxin-like ADP-ribosyltransferases (ARTDs; previously referred to as poly-ADP–ribose polymerases or PARPs) and the removal of ADPr is catalyzed by a family of glycohydrolases. Although thousands of potential ADPr modification sites have been identified using high-throughput mass-spectrometry, relatively little is known about the sequence specificity encoded near the modification site. Herein, we present a matrix-assisted laser desorption/ionization time-of-flight (MALDI-TOF) method that facilitates the *in vitro* analysis of proximal factors that guide ARTD target selection. We identify a minimal 5-mer peptide sequence that is necessary and sufficient to drive glutamate/aspartate targeting using PARP14 while highlighting the importance of the adjacent residues in PARP14 targeting. We measure the stability of the resultant ester bond and show that non-enzymatic removal is pH and temperature dependent, sequence independent, and occurs within hours. Finally, we use the ADPr–peptides to highlight differential activities within the glycohydrolase family and their sequence preferences. Our results highlight (1) the utility of MALDI-TOF in analyzing proximal ARTD–substrate interactions and (2) the importance of peptide sequences in governing ADPr transfer and removal.

## 1. Introduction

ADP-ribosylation is a ubiquitous post-translational modification (PTM) found in a vast array of species [1]. Despite being one of the first characterized PTMs [2], the biochemical mechanisms governing ADP-ribosylation are still not fully understood. In humans, the transfer of ADP–ribose(ADPr) from nicotinamide adenine dinucleotide (NAD^+^) to the protein target (Figure 1a) is mediated by a family of seventeen diphtheria toxin-like ADP-ribosyltransferases (ARTDs) that were previously referred to as poly-ADP–ribose polymerases (or PARPs) [3]. While the enzyme family is collectively referred to as ARTDs, the individual proteins are still identified based on their PARP numbering, a system we will adopt herein to refer to specific enzymes [4]. The ARTD family is further subdivided based on whether a single ADPr unit is transferred (PARP3–4, 6–8, 10–12, and 14–16) or whether the initial ADPr can be elongated with multiple ADPr units (PARP1–2, TNKS1–2) [5]. Initially discovered as DNA base repair enzymes based on the activity of PARP1 in the nucleus [6], the ARTD family has since been linked to a growing set of biological pathways and disease states [7,8]. The removal of ADPr is catalyzed by a separate class of glycohydrolases [9], which can be subdivided based on their ability to remove poly-ADP–ribose (PAR, PARG) [10], mono-ADP–ribose (MAR; *macro*D1, *macro*D2, and ARH1) [11,12,13], or both (TARG1, ARH3) [12,14,15]. As with the ARTD family, ADPr erasers have been implicated in a number of essential biological processes [9]. 

A fundamental challenge in understanding the function of MAR/PAR dependent signaling is elucidating the mechanisms of ARTD target selection. Target selection can be broadly attributed to two types of intermolecular interactions: (1) The distal interactions between the various domains attached to the catalytic fragment that help recruit and hold substrates in place for modification (such distal interactions between substrate and/or allosteric regulators can also be required to fully activate the enzymes *in vivo* as seen with PARP1 and 2 during DNA damage repair [16,17]); and (2) the proximal interactions between the catalytic domain and the substrate itself. While the investigation of both types of interaction (proximal and distal) is required to fully understand how ARTDs identify and modify their cellular targets, the determination of specific factors that drive proximal modification would be a useful complement to ongoing efforts to identify target sequences using tandem mass-spectrometry (MS/MS) [18,19]. 

As MS/MS based efforts have vastly expanded our understanding of the ADP-ribosylome by revealing thousands of potential MAR/PAR sites in the human proteome, the task of identifying bona fide sites has become even more daunting. Bioinformatic analyses of the resultant data have identified putative site motifs [20], but those sites are often weakly enriched and not fully validated. The lack of a sequence motif could be due to: (1) A real lack of a consensus for a target sequence within the ARTD family; (2) the presence of weak motif signatures within an otherwise promiscuous enzyme class; and/or (3) the artificial enrichment of non-physiological sites due to the various interventions required to enrich MAR/PAR sites in MS/MS procedures. Understanding the proximal factors that influence ARTD target selection will help distinguish between these three possibilities. Additionally, the same factors that dictate ADPr transfer could impact ADPr hydrolysis, thereby expanding the importance of sequence motifs more broadly to the glycohydrolases. 

Ultra-thin layer chromatography matrix-assisted laser-desorption/ionization time-of-flight (TLC-MADLI) analysis provides a unique opportunity to interrogate proximal ARTD–substrate interactions. Unlike MS/MS, where it can be difficult and/or expensive to determine the relative levels of MAR/PAR for each identified site, TLC-MALDI allows for the direct comparison of specific sequences and the quantification of their ADP-ribosylation *in vitro*. The pre-application of a thin-layer of matrix on the steel objective enhances crystal formation in the presence of common contaminants and improves the resultant signal intensity and resolution [21,22,23]. Further, the ability to rapidly and inexpensively probe peptide substrates in isolation facilitates the characterization of each amino acid’s contribution to ARTD activity without the confounding input from multiple MAR/PAR sites found on whole proteins. By applying TLC-MALDI analysis to peptides identified through MS/MS screening methods, experimentalists can make detailed assessments of the relative contribution of each sequence to ADP–ribose activity, providing a complementary method to identify bona fide MAR/PAR sites. Each of these features of TLC-MALDI is vital for uncovering proximal preferences that guide ADP-ribosylation.

We previously used TLC-MALDI to identify an 18-mer peptide (P14p1) found on PARP13 that was preferentially labeled by the catalytic fragment of PARP14 (hereafter referred to as P14c) *in vitro* [24]. Herein, we describe the expansion of our efforts to quantitatively assess the proximal preferences of P14c for this peptide (Figure 1b). We demonstrate that a truncated five amino acid sequence is sufficient to drive modification via P14c and we identify the positions within that sequence that are required for maximal ADPr transfer. We develop a three-step purification strategy to chemoenzymatically synthesize homogenous MARylated peptides using our findings. We use the resultant MAR-peptides to assess ADPr removal and describe the effects of specific sequences on both enzymatic and non-enzymatic hydrolysis. We observe that the non-enzymatic hydrolysis of MAR occurs within hours in mild conditions, a result which highlights the potential reversibility of ADP-ribosylation *in vivo* and impacts future site identification methods. Taken together, our findings elucidate proximal preferences for P14 for these peptidyl substrates and demonstrate the wider importance of proximal sequences in ADPr-dependent signaling.

## 2. Results and Discussion

Initially, a series of P14p1 truncations were designed (P14p2-6) to identify the necessary and sufficient sequence elements required for P14c preferential modification (Figure 2a, full sequences in Appendix A). Each of the peptides were incubated in the presence of P14c and NAD^+^ and subjected to TLC-MALDI. All of the resulting spectra displayed varying levels of modification by MAR as indicated by the expected mass shift of +541 Da (Figure 1a and Figure 2b). Integration of the unmodified and modified peaks in the MS spectra was used to quantify the relative levels of ADP-ribosylation for each of the truncated peptides (Figure 2c). Dividing the integrated area of the modified peptide by the total area of both the modified and unmodified peptide allowed for the determination of the total percentage of modified peptide produced in the experiment. The loss of C-terminal residues to the modified glutamate (E) resulted in a decrease in ADPr transfer, though this effect is not significant until the acceptor residue is lost (P14p3), leaving behind a tyrosine (Y) or a number of serines (S) as potential acceptors. These data are consistent with recent studies identifying Y as a modified site on PARP14, but the ARTD responsible for modification of Y on PARP14 is still unknown [25]. However, it appears that neither Y nor the other residues are the preferred acceptor sites for P14c within this sequence, as their presence as alternative P14c targets resulted in a nearly three-fold decrease in modification (from 16.7% to 6.2% of the total peptide). By contrast, the N-terminal portion of the 18-mer can be truncated to an overlap of two amino acids with no apparent loss in activity (compare P14p1 to P14p5). While a minor contribution of the final three C-terminal residues in P14c targeting cannot be ruled out, a truncated five amino acid sequence surrounding the acceptor E residue showed no significant loss in P14c activity.

As the catalytic fragment of P14c is likely more promiscuous than the full-length protein, it was important to confirm that the trends observed with P14c were representative of the native protein. Previous work by Barbarulo and colleagues showed that a longer construct (with a 553 amino acid N-terminal truncation of P14) complemented the activity of full-length P14 in a P14 knock-out model [26]. This nearly full-length construct (P14-∆553) is compatible with bacterial expression and was selected as a proxy for full-length activity. While the level of modification was lower overall with P14-∆553 than that observed with P14c, a nearly identical pattern of ADP-ribosylation occurred with the tested peptides (Figure 2d, representative spectra in Appendix A). These data demonstrate that, while P14c is possibly more promiscuous than longer forms of P14, the proximal factors that guide P14 targeting are recapitulated in the shorter construct. Given the higher activity of P14c, its ability to be expressed and purified more readily in bacteria, and its utility as a proxy for P14 full-length activity, it was selected as the primary construct going forward. Importantly, the incubation of any of the tested peptides with PARP15, a closely related P14 ortholog, resulted in almost no transfer of ADPr (Appendix A). Given the low levels of MARylation in the presence of P15, the activity of P15 was confirmed using an auto-modification assay (Appendix A). As P15 and P14 are closer orthologs than the other ARTD family-members, it is interesting to note that they do not share a preference for the same minimal peptide sequence. While determining if the remaining ARTD family members are active in the presence of this minimal sequence is a motivating topic for future investigation, these findings nonetheless demonstrate that the 5-mer sequence is preferentially targeted by P14c and will be an optimal minimal fragment to assess the effects of proximal sequence on P14 activity.

Next, we assessed the relative contribution of each of the non-acceptor amino acids on P14 selection. We systematically replaced each of the residues surrounding the E acceptor (and the acceptor itself) with alanine (A) and performed TLC-MALDI with P14c as described above (Figure 3a). Our initial experiments with peptide variants that involved the substitution of a charged residue with alanine resulted in spectra that had severely diminished signal intensities compared to the parent peptide. As ionization in the positive mode used in our MALDI-TOF is dependent on the overall charge of the molecule, we theorized that this lack of signal was caused by the loss of a +1 charge on these peptides. To avoid artifacts due to charge imbalances in the peptides tested, we added back the lost residue to the C-terminus of the peptide while maintaining the alanine substitution (e.g., P14p8 and P14p9). This allowed us to maintain the same charge on each peptide and allowed for direct comparisons in spectral intensity across the screened mutants (Figure 3b). Replacing the glutamate acceptor with alanine resulted in an expected significant decrease in MARylation (3.5% versus 12.2%), confirming this as the primary acceptor site (Figure 3c). The 3.5% modification rate is therefore the background level of non-preferential transfer in this *in vitro* setting. A decrease in MARylation similarly occurred when the acceptor minus two (P14p8) and minus one (P14p9) positions were altered to alanine as compared to the native sequences in P14p6 (9.3% or 7.6% peptide modification versus 12.2%, respectively). However, the change with P14p8 was not significant. As both histidine (H) and arginine (R) are positively charged, these data suggest that P14 prefers either a basic and/or larger residue in the two N-terminal positions to the acceptor residue. However, as there was no change in ADPr transfer with an alanine substitution at the two C-terminal positions to the acceptor residue, they are non-essential for P14 targeting.

P14 is thought to modify aspartate (D) as well as E, so a final swap from E to D in the 5-mer was similarly tested for P14c activity. There was a non-significant decrease (8.9%) in labeling when D was the primary acceptor, confirming the ability of P14c to modify both D and E within these minimal peptide substrates. The activity of P14-∆553 was also assessed to confirm that the significant decreases in labeling observed with P14p9 and P14p10 were not attributable to a promiscuity difference in P14c. For both peptides, there was no detectable labeling in the presence of P14-∆553 (Appendix A). These results match the observations made with P14c and confirm that both the catalytic fragment and the longer PARP14 proteins share a similar set of proximal preferences when identifying a target sequence *in vitro*.

To determine whether the 37.7% decrease in P14c activity we observed for the P14p9 peptide was due to the positive charge in the arginine (R) or its larger size compared to A, two new peptide substrates were designed with either methionine (M, P14p13) or lysine (K, P14p14) at the acceptor minus one site (Figure 3d). As with the A substitution, the presence of the larger, though uncharged, M residue results in a 44% decrease in P14 activity. Therefore, the decrease observed with an A substitution was likely not due to the difference in size at this position. The K substitution—which maintains its charge at this position—resulted in a 46% increase in P14 activity. It appears that, while R is found at this position in the original sequence, a K residue further enhances activity. Interestingly, the preference for a lysine in the acceptor minus one site has been observed for PARP1 [27] and was one of the putative motifs suggested in prior proteomic analyses of the ADP-ribosylome [20]. 

To analyze the potential requirement for a minimum of two C-terminal residues for ADP–ribose transfer, a set of peptides were designed with the R-E sequence slid towards the C-terminus (P14p15 and P14p16) (Figure 3g). In both cases, the loss of amino acids C-terminal to the E acceptor resulted in a non-significant decrease in overall ADP-ribosylation (8.2% for both P14p15 and P14p16 versus 12.2% for P14p6, Figure 3h,i). Therefore, while a minimal overhang is required for maximal transfer, it is not as essential as the preference for a K/R in the position N-Terminal for the acceptor residue. Taken together, these results demonstrate a general promiscuity in P14c targeting and validate a preference for sequences with a basic–K/R–D/E–X–X signature.

Following the identification of a putative motif for P14-dependent ADPr transfer, we determined whether this motif influenced enzymatic and non-enzymatic MAR hydrolysis. Like other ester bonds, the peptide–MAR linkage is base labile [28]. However, previous studies on the reversal of the MAR ester bond routinely used basic conditions well above the pH range observed in the cell (pH > 9) [5,29], making it difficult to interpret how stable this modification is *in vivo*. The presence of multiple MAR/PAR sites on proteins further complicates the analysis of single-site hydrolysis. We reasoned that our TLC-MALDI approach would be well suited to studying the kinetics of single-site MAR removal without these complications. P14c was used to label both P14p6 and P14p9 homogenously, and MARylated peptide was purified using a three-step method. The resultant P14p6–MAR peptide was equilibrated in a range of pH conditions (pH 6–8) and the removal kinetics at 37 °C were monitored using TLC-MALDI (Figure 4a). The hydrolysis of ADP–ribose was best described by a pseudo first-order reaction, which was used to determine the half-life of MAR hydrolysis. The half-life of MAR–peptide at pH 8 is 2.2 h and is only slightly longer at pH 7 (t_1/2_ = 6.0 h) (Figure 4b). Only by equilibrating the MAR–peptide in slightly acidic conditions were half-lives in the day range (t_1/2_ = 20.4 h) observed. The hydrolysis assays were repeated at room temperature (25 °C) to investigate the effect of temperature on the stability of the MAR–peptide bond. Predictably, lowering the temperature slowed the loss of MAR at all pHs tested, though there is still appreciable removal within the mild base (t_1/2_ = 5.6 h at a pH of 8) (Figure 4c). However, equilibration in slightly acidic conditions significantly slows the hydrolysis of MAR and results in a half-life of greater than two days (Figure 4d). Lowering the temperature even further to 4 °C stabilized the modification, though it did not prevent hydrolysis at either pH 8 or 7 (Figure 4e). Of all the conditions tested, only the equilibration of MAR–P14p6 in acidic conditions on ice seemed to halt non-enzymatic removal (Figure 4f). These results highlight the dynamics of ADP-ribosylation *in vitro* and have important implications for the methodologies utilized to study ADP-ribosylation (as discussed further below).

Next, we wanted to determine whether the observed specificity was attributable to either (1) a specific interaction between the acceptor minus one site and P14, or (2) a stabilizing intramolecular interaction between the positive charge on the adjacent R residue and MAR that slows hydrolysis. To discern this, hydrolysis assays were performed with P14p9, and the rates of hydrolysis for P14p9 were compared to those for P14p6 (Appendix A). No significant difference was observed in any of the tested conditions between the non-enzymatic rates of hydrolysis for P14p6 and P14p9. As such, the observed increase in MARylation activity seen for P14p6 is likely due to a specific interaction with P14 rather than its susceptibility to hydrolysis.

After surveying non-enzymatic hydrolysis, we moved on to study the effects of proximal sequence on glycohydrolase activity. We limited our study to removal enzymes that had previously been validated as being both D/E and mono-ADPr selective: *macro*D1, *macro*D2, and TARG1 [11,12,13,14,15]. Each of the glycohydrolases were incubated with either P14p6 or P14p9, and the amount of MAR hydrolysis present was compared to a non-enzymatic control reaction (Figure 5a). All three of the erasers showed activity for the P14p6 peptide, with *macro*D2 displaying a 3.8-fold increase in activity compared to *macro*D1 and a 2.0-fold increase compared to TARG1 (Figure 5b). *Macro*D2 is, therefore, the most active enzyme in our experimental conditions with peptidyl substrates; however, it is important to note that the effects of removal on full-length proteins in a cell-based context requires further investigation to determine how they differ from *in vitro* conditions with a simple peptide. 

Next, each of the erasers was incubated with the P14p9 peptide and MAR removal was assessed to determine the effects of the acceptor minus one position on glycohydrolase activity. As with P14p6, each of the enzymes tested was active in the presence of the MARylated P14p9 peptide (Figure 5a). However, differential sensitivities were observed for the substitution at the acceptor adjacent position, with *macro*D1 displaying no significant difference in activity for P14p6 versus P14p9, while both *macro*D2 and TARG1 had a 30–40% decrease in activity with the altered sequence (Figure 5b). While potential differences between the activities of the glycohydrolases *in vitro* and in a cellular context could account for some of this discrepancy, it is interesting to find a relative difference in sequence preference at the proximal site for these erasers. Taken together, this investigation of glycohydrolase activity with P14c generated MAR–peptides reveals differences in substrate preference and sensitivity to proximal sequences while demonstrating the utility of TLC-MALDI in elucidating fundamental ADP–ribose biochemistry.

## 3. Materials and Methods

### 3.1. Materials

Recombinant proteins were preceded by either a 6x His tag in a pET-28b vector (P14c, P15c, or *macro*D1), a GST tag in a pGEX-6P-2 vector (*macro*D2 or TARG1), or a 6x His, SUMO tandem tag in a pET-28b vector (P14-∆553) and were expressed and purified as previously described [30]. All proteins were derived from human variants and their Uniprot identifiers are Q460N5 (PARP14), Q460N3 (PARP15), Q9BQ69 (mD1), A1Z1Q3 (mD2), and Q8IXB3 (TARG1). The full proteins for the glycohydrolases were inserted into their respective cloning vector, with residues 1459–1801 used for P14c, residues 481–678 used for P15c, and residues 553–1801 used for P14-∆553. Expression vectors were sequenced to confirm DNA insertion and protein concentrations were determined using a Bradford assay. As there were bacterial proteins co-purified with both P14-∆553 and *macro*D1, their concentrations were determined using a gel-based assay with BSA as a standard. A representative Coomassie stained gel loaded with 500 µM of each enzyme used in this study can be found in Appendix A. Peptides lacking a Y or W residue had a W added to the C-terminus to aid in concentration determination. The peptides were obtained from GenScript.

### 3.2. Methods

#### 3.2.1. ADP-Ribosylation Assay

P14c or P15c (5 µM) was incubated with 500 µM NAD^+^ (Sigma-Aldrich, St. Louis, MO, USA) and the indicated peptide (10 µM) for 10 min at 30 °C in a 16 µL reaction volume consisting of 25 mM HEPES, pH 7.5, 12 mM MgCl_2_, 50 mM NaCl, and 0.5 mM TCEP. Reactions were quenched by the addition of 16 µL of 0.1% TFA. 2 µL of sample was mixed with 4 µL of TLC-MALDI buffer (MB, 5 mg/mL α-cyano-4-hydroxycinnamic acid (CHCA, Sigma-Aldrich), 25% acetonitrile, 0.05% TFA, and 5 mM NH_3_PO_4_) and subjected to TLC-MALDI analysis. P14-∆553 was treated as described above except the final concentration was set to 2.5 µM and peptide concentration was 5 µM due to the lower yield of the full-length protein. For the P15c automodification assay, 2 µL of the reaction mixture was mixed with 2 µL of 2× sample buffer, and substrate labeling was detected via immunoblot using the pan-ADP–ribose detection reagent MABE1016 (Millipore, Hayward, CA, USA), quantified using ImageLab v5.2 (Bio-Rad, Hercules, CA, USA), and compared against the total substrate load as determined using Ponceau S stain.

#### 3.2.2. Non-Enzymatic Hydrolysis Assay

Synthesized MAR–P14p6 or –P14p8 was equilibrated to 50 µM in 10 mM potassium phosphate, pH 6.0 and diluted in 50 µM BIS-Tris, pH 6.0, 7.0, or 8.0 to a final concentration of 10 µM. MAR–peptides (2 µL) were incubated at either 37 °C or 25 °C and samples were quenched at 0, 45, 90, 180, or 300 min in 4 µL of MB and subjected to TLC-MALDI analysis. For the 4 °C incubations, samples were collected at 0, 24, 48, and 72 h.

#### 3.2.3. Glycohydrolase Assay

MAR–P14p6 or –P14p8 (10 µM) was incubated with either *macro*D1, *macro*D2, or TARG1 (500 nM) for 10 min at 30 °C in 2 µL reaction volume consisting of 25 mM BIS-Tris, pH 7.0, 50 mM NaCl, and 0.5 mM TCEP. Reactions were quenched with 4 μL of MB and subjected to TLC-MALDI analysis.

#### 3.2.4. Chemoenzymatic Synthesis of MAR–Peptide

P14 (8 µM) was incubated with 4 mM NAD^+^ and either P14p6 or P14p8 (115 µM) for 2 h at room temperature in a 420 µL reaction volume consisting of 50 mM HEPES, pH 7.5, 400 mM NaCl, 2.5% glycerol, and 0.4 mM β-mercaptoethanol. At 30 min an additional 11.8 µL of concentrated P14 (285 µM) was added. The reaction was diluted to 1 mL in an SE buffer (10 mM potassium phosphate, pH 6.0 and 25 mM NaCl) and subjected to size exclusion chromatography at a rate of 0.75 mL/min using a Superdex Peptide 10/300 GL column (Cytiva, Marlborough, MA, USA) while collecting 250 µL fractions. MAR–peptide-containing fractions were pooled and brought up to 5 mL in an IE buffer (10 mM BIS-Tris, pH 5.0 and 25 mM NaCl) and further purified using anion exchange chromatography with a HiTrap Q HP column (Cytiva) and exchanged into an IE buffer + 1 M NaCl. ADPr–peptide fractions were desalted using a Sep-Pak C18 cartridge (Waters, Milford, CT, USA). Desalted peptides were diluted 1:1 with 100% acetonitrile, flash-frozen in liquid nitrogen, and lyophilized. Samples were stored at −80 °C immediately following drying to prevent ADPr hydrolysis.

#### 3.2.5. TLC-MALDI Preparation and Sample Cleanup

The ultra-thin layer was prepared as previously described [24]. 2 µL of sample in MB was spotted directly on the thin-layer and dried. After drying, spots were washed twice with 5 µL of ice-cold 0.1% TFA for 1 min. Calibrations were performed using Peptide Calibration Standard II (Bruker, Billerica, MA, USA) per the manufacturer’s instructions. MALDI-TOF experiments were performed using a MALDI-8020 instrument (Shimadzu, Kyoto, Japan).

#### 3.2.6. MS Acquisition Parameters and Data Analysis

The collection of peptide spectra was performed using an automated firing method defined by a regular circle with a TV Raster (1500 µm diameter, 50 points, no offsets, 199 µm spacing) and random dithering (250 ms dwell time, 20 µm radius) with a mass range set to 700–2800 Da. Fifty laser shots were fired for each profile at a frequency of 50 Hz, and 2500 total shots were collected per sample. Post-acquisition baseline subtraction and smoothing were performed using MALDI Solutions (Shimadzu) with the following parameters: baseline filter width set between 15–30, Savitsky–Golay smoothing with a smoothing width between 5–30, and peak width set to 5. The peak delimiter method was set to threshold apex with a threshold of 0.01 mV and exclusion of peaks with intensities less than 0.01 mV. Following data acquisition, selected unmodified and ADPr-modified peaks were identified based on their expected mass and integrated to determine the area under the curve using MALDI solutions The resulting values were used to calculate relative levels of ADP-ribosylation. The percent peptide modification was calculated by dividing the area from the modified peak by the sum of the modified peak and the unmodified peak. The center of each peak was recorded, and the spectra were rejected if any observed mass differed from the predicted mass by more than 1%. The spectral intensities were min-maxed normalized between 0.0 and 0.1 in Origin 2020 (OriginLab, Northampton, MA, USA) and centered on the unmodified peak to aid comparisons between different peptides. Student’s *t*-tests were performed with a two-tailed distribution to determine significance for any observed differences between experimental conditions in Excel Professional Plus 2016 (Microsoft, Redmond, WA, USA). Linear fitting of non-enzymatic MAR hydrolysis to a pseudo first-order rate expression was performed in Origin 2020.

## 4. Conclusions

Herein, we demonstrated the applicability of the TLC-MALDI approach in the identification and validation of site motifs within the ARTD family. Working with a minimal motif containing a single acceptor site facilitated the determination of the effects of the surrounding amino acids on the activity of P14c. While P14c is broadly promiscuous, there is a distinct preference *in vitro* for sites with a basic–K/R–D/E signature. As we have presented a method for generating ADP-ribosylated peptides, it will be interesting to include these reagents in structure-based efforts to determine how the proximal basic–K/R signature influences ADP–ribose transfer. It will be important to discover if the proximal sequence directly interacts with P14 or if it plays a more indirect role in stabilizing the negative charge present within the attached phosphodiester. Based on our prior work with P14, we were also curious how prevalent this signature was within previously identified P14 targets. Unfortunately, the number of validated P14 specific target sites is still limited (eight sites total on PARP13). Of those sites, a single site has this specific signature, though this was the site that was modified the most rapidly in the prior study [31]. There is a clear need for further work identifying ARTD family-member specific targets, and it will be interesting to compare how the *in vitro* preferences found herein can be compared to future modification sites identified in ongoing proteomic efforts. It will also be worthwhile to apply this technique to the remaining ARTD family members with their own unique target sequences. Completion of this comparative analysis will help reveal the level of specificity within the family and build on the mechanism for ARTD targeting. Regardless, our current results can immediately be used to screen putative sites from proteomic surveys to identify bona fide *in vivo* P14 sites. 

It is important to note that there are inherent limitations to the work we have described herein. Our study was designed to explore the minimal motifs required to guide proximal targeting by PARP14, which means we were only considering linear sequences and not considering the effects of folded proteins on targeting. It is certainly possible that the folded nature of various sites could influence their accessibility at the PARP14 active site. However, the recently proposed structure of PARP13 on AlphaFold has localized the P14p1 site to a disordered loop, which would be similar in structure to our analyzed peptides [32,33]. We also focused our study primarily on a single peptide. While we would argue that we have selected one of the more compelling PARP14 sites based on the previous literature, it will be important to explore how the proximal interactions we have identified are shared between future PARP14 sites and those found for other ARTD family members. Finally, P14c and the longer PARP14 variants undergo significant MARylation both *in vitro* and *in vivo* [25,31]. The presence of MARylation on these proteins could certainly alter target selection, and it would be interesting to compare how proximal preferences are impacted by the loss of auto-modification sites.

One of the main advantages of the described work is the ability to interrogate both the attachment and removal of MAR at a single minimal acceptor site. TLC-MALDI reveals that the esterification of peptide with MAR results in a far more labile modification than previously described [5,29]. While certain factors in the cell (e.g., burying of sites within protein–protein interaction surfaces, intracellular compartmentalization, etc.) could certainly alter the hydrolysis dynamics we have uncovered, understanding how the innate stability of MAR at D/E residues impacts signal persistence will be of particular interest going forward. Moreover, recent studies have shown that MAR/PAR transfer is not restricted to a single type of amino acid, with PARP activity observed on arginine (R), lysine (K), cysteine (C), histidine (H), serine (S), tyrosine (Y), and the aforementioned D/E [25,34,35,36,37,38]. Expanding this technique to additional types of MAR/PAR–peptide bond chemistries will help uncover how the unique chemistry at the acceptor site affects PTM lability. Investigation of the interplay between site chemistry and MAR/PAR stability could have important implications for how the kinetics of non-enzymatic removal impact MAR/PAR signal transduction pathways. Recent work completed by Tashiro and colleagues demonstrated a similar effect of non-enzymatic hydrolysis on synthetically derived MARylated peptides using an orthogonal HPLC method [39]. Combined with our findings, these data suggest that esterification of acidic residues by ADP–ribose is a highly reversible process in mild conditions. These results similarly hint at a larger potential pool of underrepresented MAR/PAR sites that have been overlooked due to the instability of the ester bond. When preparing protein libraries for MS/MS analysis, it is fairly common to equilibrate the digests in pH neutral conditions at elevated temperatures for more than a day (e.g., overnight treatment with trypsin). Based on the current findings, it is possible that treatment of peptide libraries in this manner would result in a significant loss of MAR/PAR from acidic acceptor sites. In a complex pool of acceptor site chemistries, this could result in the over-enrichment of non-ester linkages and an apparent absence of D/E modifications. Therefore, the exploration of the impact of temperature and pH on MS/MS-based site discovery has the potential to identify an underrepresented population of sites.

Finally, the TLC-MALDI method has been successfully adapted from quantifying ADPr transfer to ADPr removal using several different glycohydrolases. As this class of enzymes were thought to be fairly substrate agnostic, it was surprising to find differential substrate preferences based on proximal interactions [39,40]. For example, *macro*D1 has been previously shown to remove ADPr from a range of substrates (e.g., DNA, RNA, and protein) and can function as an O-acetyl-ADPr deacetylase [41]. The data support the role of *macro*D1 as a promiscuous removal enzyme, but this lack of specificity might contribute to a lower activity for the peptidyl substrates analyzed. Recent work by Žaja and co-workers identified an enrichment of *macro*D2 in neuronal cells [42]. Further, similar efforts have highlighted specific neurological phenotypes in mice models associated with a loss of *macro*D2 [43]. Coupled with the observation that *macro*D2 is the most active eraser of the P14c generated MAR–peptides, this could hint at a potential antagonistic role for P14 and *macro*D2 in the brain. Obviously, it is not possible in this study to definitively state that any of the tested glycohydrolases are P14 specific in the cell. Further work will be required to determine if the preferences observed *in vitro* with peptidyl substrates will match the *in vivo* preferences with whole proteins. Combined, these results have expanded the role TLC-MALDI can have in analyzing ADP-ribosylation while providing new insights into the mechanism of target selection, and we suggest this as a complement to ongoing efforts to examine the function of ADPr-dependent signaling. 

## Figures and Tables

**Figure 1 molecules-28-06061-f001:**
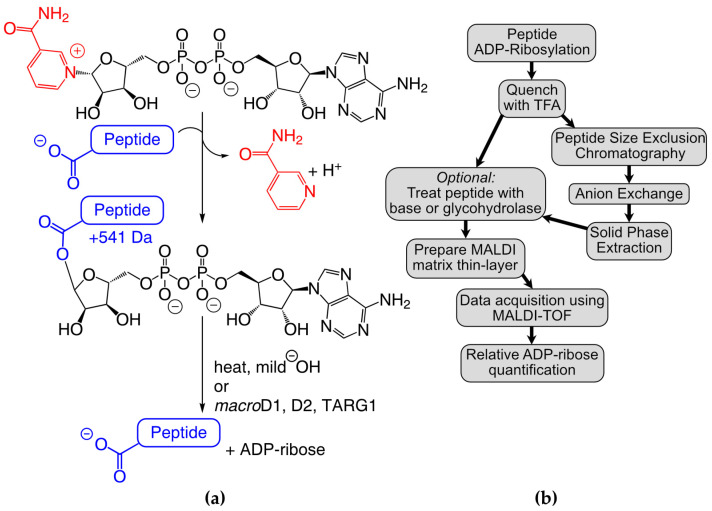
Overview of the experimental approach. (**a**) ADP-ribosylation (step 1) and hydrolysis reactions (step 2). (**b**) TLC-MALDI workflow.

**Figure 2 molecules-28-06061-f002:**
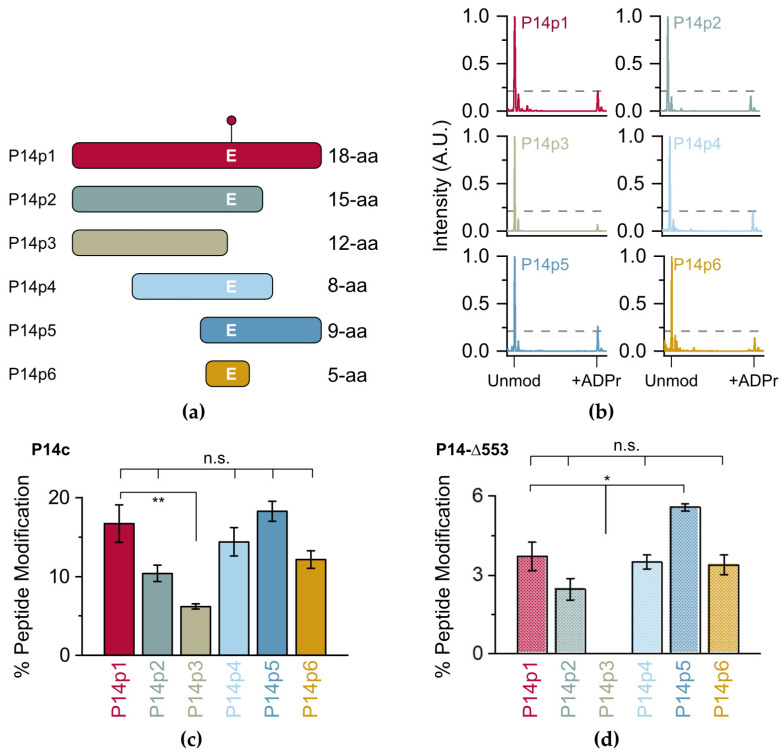
Identification of a minimal P14 selective peptide sequence. (**a**) Peptide truncations used in this study (full sequences in Appendix A). The E acceptor is indicated. (**b**) P14c and the indicated peptide were incubated in the presence of NAD^+^ and subjected to TLC-MALDI to visualize the resulting increase in m/z due to MARylation (+541 Da). The dashed line represents the intensity observed for ADP-ribosylation of P14p1. (**c**) MS spectra for P14c treatment were integrated to determine the relative levels of ADP-ribosylation. The bar graphs depict the fraction of the total peptide that was modified (mean ± S.E.M., n = 3). (**d**) MS spectra from P14-∆553 treatment were integrated as in (**c**). The bar graphs depict the fraction of the total peptide that was modified (mean ± S.E.M., n = 3). * represents *p*-value < 0.05 and ** represents *p*-value < 0.01, two-tailed Student’s *t* test, n.s. represents a non-significant difference.

**Figure 3 molecules-28-06061-f003:**
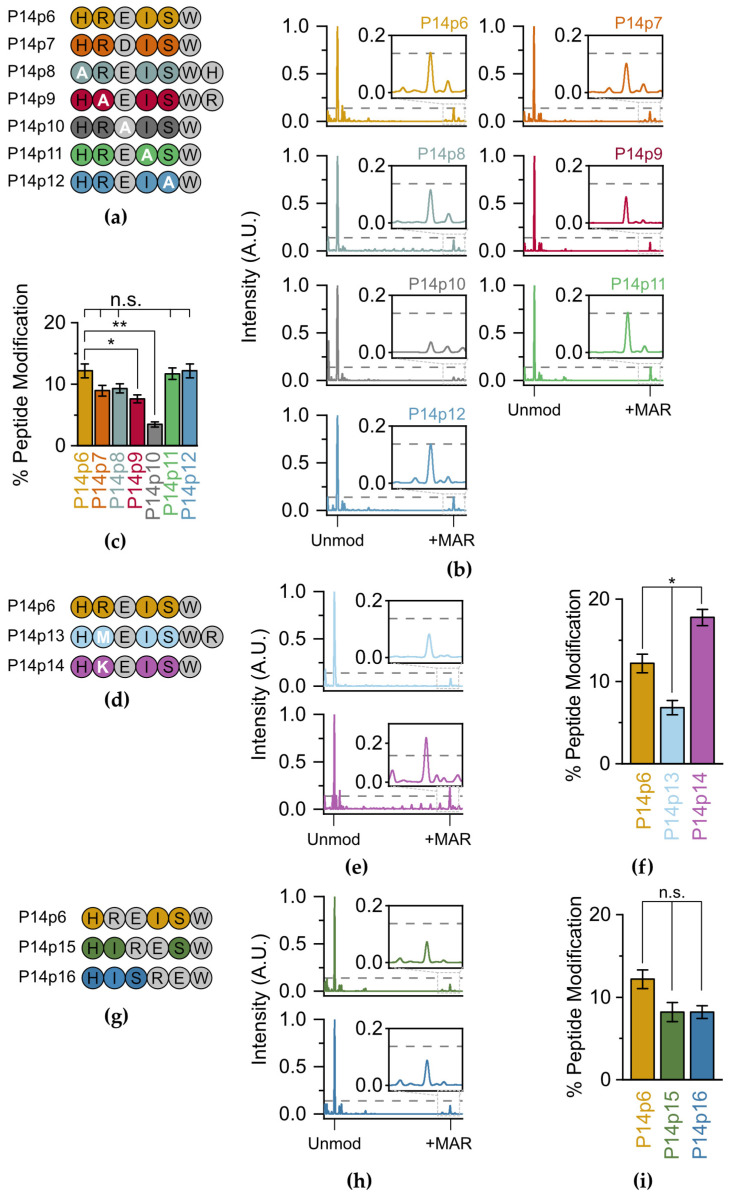
P14 preferentially MARylates a basic–K/R–D/E proximal sequence. (**a**) Alanine (A) substituted peptides used in this study. (**b**) P14c and the indicated peptide were incubated in the presence of NAD^+^ and subjected to TLC-MALDI to visualize the resulting increase in *m*/*z* due to MARylation (+541 Da). The dashed line represents the intensity observed for ADP-ribosylation of P14p6 and the inset highlights the +MAR spectra. (**c**) MS spectra were integrated to determine the relative levels of ADP-ribosylation. The bar graphs depict the fraction of the total peptide that was modified (mean ± S.E.M., n = 3). * represents *p*-value < 0.05 and ** represents *p*-value < 0.01, two-tailed Student’s *t* test, n.s. represents a non-significant difference. (**d**) Methionine (M) and lysine (K) substituted peptides used in this study. (**e**) P14p13 and P14p14 MS spectra normalized as in (**b**). (**f**) Relative levels of P14p13 and P14 p12 MARylation analyzed as in (**c**). (**g**) Shuffled peptides with the R-E signature slid towards the C-terminus. (**h**) P14p15 and P14p16 MS spectra normalized as in (**b**). (**i**) Relative levels of P14p15 and P14p16 MARylation analyzed as in (**c**).

**Figure 4 molecules-28-06061-f004:**
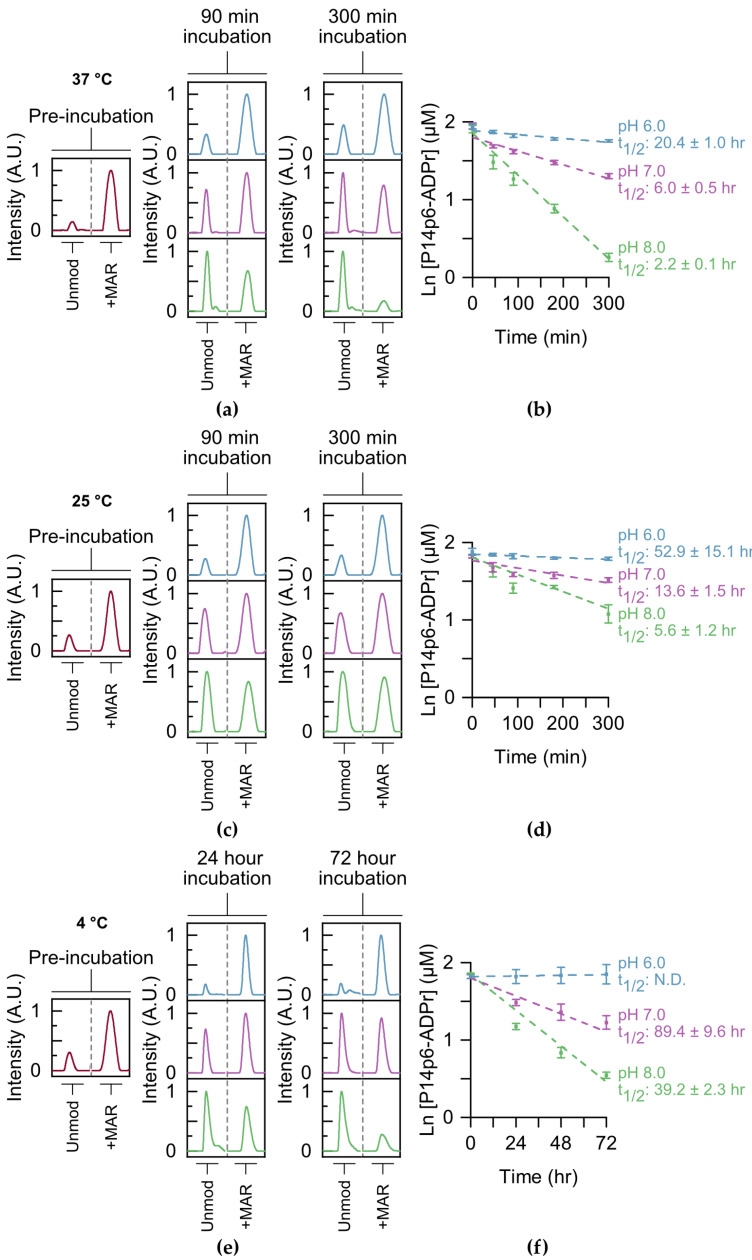
The MAR–peptide bond is hydrolyzed under mild conditions. (**a**) A synthesized MAR–P14p6 peptide was equilibrated at 37 °C, and pH 6.0 (blue), pH 7.0 (purple), or pH 8.0 (green) and MAR hydrolysis was monitored at the indicated times using TLC-MALDI. The unmodified and modified peaks are shown for comparison. (**b**) MS spectra were integrated to determine the relative levels of MAR hydrolysis and fit to a pseudo first-order rate expression to determine the half-life of the modification (mean ± S.E.M., n = 3). (**c**) Incubation at room temperature stabilized the MAR–peptide bond. Experiments were performed as in (**a**) at 25 °C. (**d**) Determination of MARylation half-lives at 25 °C. (**e**) Incubation on ice effectively halts MAR hydrolysis in mild acid. Experiments were performed as in (**a**) at 4 °C. (**f**) Determination of MARylation half-lives at 4 °C.

**Figure 5 molecules-28-06061-f005:**
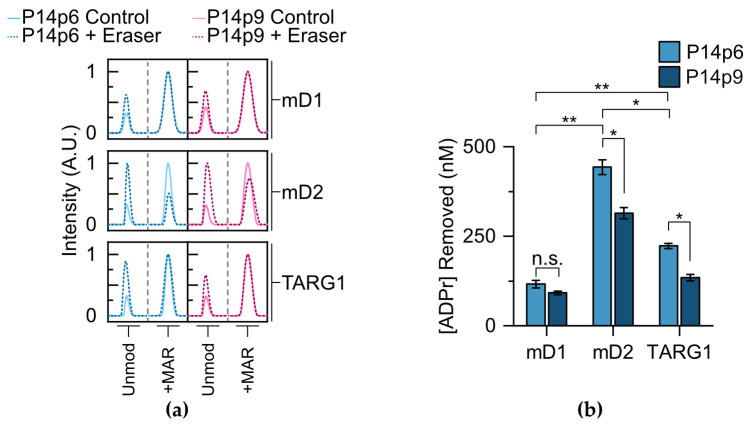
ADPr glycohydrolases display differential preferences for P14 selective sequence motifs. (**a**) Synthesized MAR–P14p6 or MAR–P14p9 peptides (dashed lines) were incubated in the presence of either *macro*D1 (mD1), *macro*D2 (mD2), or TARG1 and subjected to TLC-MALDI to determine the relative levels of hydrolysis. Non-enzymatic controls (solid line) were utilized to ascertain the levels of background hydrolysis. The unmodified and modified peaks are shown for comparison. (**b**) MS spectra were integrated to determine the relative levels of MAR hydrolysis. The bar graphs depict the amount of MAR removal (mean ± S.E.M., n = 3). ** represents *p*-value < 0.01, two-tailed Student’s *t* test, * represents a *p*-value < 0.05, and n.s. represents a non-significant difference.

## Data Availability

Not applicable.

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
