# Peer review of "Using TLC-MALDI-TOF to Interrogate In Vitro Peptidyl Proximal Preferences of PARP14 and Glycohydrolase Specificity"

_molecules, 2023, doi:10.3390/molecules28166061_

Round 1
Reviewer 1 Report
Authors explore in the article peptides that can be modified by PARP14, study the stability of the glutamate-mono-ADP-ribosylation and test tge enzymatic hydrolysis by macrodomains previously reported to hydrolyse this type of modification. The study is limited to peptides and some conclusions can be seen as an overinterpretation in terms of the available data. The focus on the peptides should be reflected in the title. Overall the manuscript is well written and easy to read. The points that should be addressed prior to publication are listed below in no particular order.
PARP is not to be used as an acronym and the authors are encouraged to use the updated nomenclature thoughout the manuscript. (PMID 34323016)
The use of ADPr, mono-ADPr, MAR, MARylation etc. could be sharpened. ADPr typically refers to free ADP-ribose and authors also define MAR for mono-ADP-ribosylation in the introduction.
Authors should mention in the abstract the residue that is modified and also elaborate the pH dependency of non-enzymatic hydrolysis.
Page 2: It could be mentioned that some interactions in the recruitment process may specifically also activate the FL enzyme (as an example PARP2 and different DNA damage models)
In the introduction it should be mentioned that the original peptide is from PARP13.
At the end of the introduction there is an overstatement "Taken together, our findings elucidate novel proximal interactions between P14 and its targets" as these interactions are not really elucidated in this manuscript. There are no studies nor discussion how this interaction happens at the molecular level, mainly just a note that the prepared peptides could be used for structural studies in the future.
It is not clear why the sequence of FL is to recombine in E. coli and why it is not possible to do codon optimization etc. to overcome this? E. coli might not be a suitable host for expression though. It would be enough to mention that a larger fragment was used and give a citation.
Some of the proteins are not of appropriate purity, but this is clearly stated in the M&M so it is ok for this purpose. It would be good to state that/if these are all human proteins and state the uniprot IDs also. It is not appropriate to call a longer PARP14 construct P14FL in the Figure S5. On page 6 conclusions cannot be directly made that the peptide preference would be the same for FL PARP14.
Please include evidence in the manuscript that the PARP15 construct is active.
The ranking of MacroD1, MacroD2 and TARG1 is not entirely possible in the experiment. What is the expected activity of the protein batches produced and precision of the concentrations etc. Typically TARG1 has low activity, and MacroD1 batch used is not pure. I would suggest the authors to simply report that MacroD2 is the most active of these three enzymes, but would not discuss the possibility of MacroD2 being the one removing PARP14 modification sites. There is not real control as a comparison point in the data to support the ranking. Authors continue with this line of thought on page 9, but this still is an overinterpretation of the data with the peptides. This can be carefully formulated in the conclusions, which is more like a discussion in this case.
In the conclusions it would be important to address the shortcoming of the study. It was done solely with linear peptides and it could indeed be that the modification sites are mostly present in loop structures and disordered regions. Also only one particular peptide was studied. It is however a full protein that is ADP-ribosylated and it has a 3D structure that could affect the recruitment as well as accessibility. PARP14 is also able to automodify itself, even just the catalytic domain alone. How would these in vitro results be interpreted in the context of the current work? Therefore, the authors should discuss the limitations of the approach also to keep the proper balance in the conclusions. It would be of interest to look at the target protein list generated by a chemical genomics study the authors refer to whether such sequences appear in the putative PARP14 target proteins and whether these would indeed be a loop region based on experimental stuctures or e.g. AlphaFold predictions.
Author Response
- “The focus on the peptides should be reflected in the title.”
Reply: We agree with the reviewer that the title should include this information. As such we have retitled the manuscript as: ‘Using TLC-MALDI-TOF to Interrogate In Vitro Peptidyl Proximal Preferences of PARP14 (ARTD8) and Glycohydrolase Specificity’.
- “PARP is not to be used as an acronym and the authors are encouraged to use the updated nomenclature thoughout the manuscript. (PMID 34323016)”
Reply: We thank the reviewer for pointing out these issues with nomenclature within our field. Based on the recommendations found within the suggested reference we have replaced any usage of PARP to describe the enzyme family with the preferred ARTD acronym. For specific usage of a particular enzyme (e.g. PARP14) we have kept the original name in accordance with the agreed upon nomenclature found in the referenced citation. For any specific proper name of an enzyme we have also included the alternate ARTD proper name. We believe that this will capture a broader audience and is in keeping with the agreed upon nomenclature.
- “The use of ADPr, mono-ADPr, MAR, MARylation etc. could be sharpened. ADPr typically refers to free ADP-ribose and authors also define MAR for mono-ADP-ribosylation in the introduction.”
Reply: We thank the reviewer for bringing this issue to our attention. We have revised the manuscript and figures to sharpen our usage of MAR, PAR, and ADPr. We believe this enhances the readability of our manuscript.
- “Authors should mention in the abstract the residue that is modified and also elaborate the pH dependency of non-enzymatic hydrolysis.”
Reply: This alteration has been completed and is now included in the revised abstract.
- “Page 2: It could be mentioned that some interactions in the recruitment process may specifically also activate the FL enzyme (as an example PARP2 and different DNA damage models).”
Reply: We agree that it is important to mention that substrate and/or allosteric interactions could alter the activity of the ARTD family of enzymes. We have now included this analysis in the introduction and have included the appropriate references.
- “In the introduction it should be mentioned that the original peptide is from PARP13.”
Reply: We have included this information in the revised introduction.
- “At the end of the introduction there is an overstatement "Taken together, our findings elucidate novel proximal interactions between P14 and its targets" as these interactions are not really elucidated in this manuscript. There are no studies nor discussion how this interaction happens at the molecular level, mainly just a note that the prepared peptides could be used for structural studies in the future.”
Reply: We thank the author for bringing this potential overstatement to our attention. We have removed this statement in the introduction and revised it with ‘Taken together, our findings elucidate proximal preferences for P14 for these peptidyl substrates,’ which we believe aligns more closely with our data.
- “It is not clear why the sequence of FL is to recombine in E. coli and why it is not possible to do codon optimization etc. to overcome this? E. coli might not be a suitable host for expression though. It would be enough to mention that a larger fragment was used and give a citation.”
Reply: We agree with the reviewer that it should be possible to codon optimize P14-FL for expression in E. coli. However, we have tried to do this very experiment and P14-FL still recombines in expression strains. The only way we have been able to stably store the P14-FL plasmid for any purpose is to transform it into NEB Stable cells. These cells are suitable for propagation of the vector, but they are not suitable for expression. We took the reviewer’s suggestion and removed language regarding the full-length construct and simply mentioned that we used the larger fragment and provided the associated citation.
- “Some of the proteins are not of appropriate purity, but this is clearly stated in the M&M so it is ok for this purpose. It would be good to state that/if these are all human proteins and state the uniprot IDs also. It is not appropriate to call a longer PARP14 construct P14FL in the Figure S5. On page 6 conclusions cannot be directly made that the peptide preference would be the same for FL PARP14.”
Reply: We were happy to include the fact that all of our constructs were derived from human sequences in the methods section. We have also included all of the associated Uniprot IDs. We corrected the reference to P14FL in Figure S5. We removed any suggestions that our results would be recapitulated by P14FL and simply said that the longer construct reflected our findings with P14c.
- “Please include evidence in the manuscript that the PARP15 construct is active.”
Reply: We agree with the reviewer that this is an important control. We have revised figure S2 to include a new panel (Figure S2b) that demonstrates P15c automodification activity using a validated anti-MAR antibody. We have included details regarding the experiment in the methods section.
- “The ranking of MacroD1, MacroD2 and TARG1 is not entirely possible in the experiment. What is the expected activity of the protein batches produced and precision of the concentrations etc. Typically TARG1 has low activity, and MacroD1 batch used is not pure. I would suggest the authors to simply report that MacroD2 is the most active of these three enzymes, but would not discuss the possibility of MacroD2 being the one removing PARP14 modification sites. There is not real control as a comparison point in the data to support the ranking. Authors continue with this line of thought on page 9, but this still is an overinterpretation of the data with the peptides. This can be carefully formulated in the conclusions, which is more like a discussion in this case.”
Reply: Once again, we think the reviewer has raised an important point regarding over-interpretation of our results. We have removed language suggesting that we could rank the actual preference of the various glycohydrolases for P14 sites. We have instead stated that ‘MacroD2 is thus the most active enzyme in our experimental conditions with peptidyl substrates.’ We reformulated our discussion in the conclusion to state ‘Coupled with the observation that macroD2 is the most active eraser of the P14c generated MAR—peptides this could hint at a potential antagonistic role for P14 and macroD2 in the brain. Obviously, it is not possible in this study to definitely state that any of the tested glycohydrolases are P14 specific in the cell. Further work will be required to determine if the preferences observed in vitro with peptidyl substrates will match the in vivo prefer-ences with whole proteins.’
- “In the conclusions it would be important to address the shortcoming of the study. It was done solely with linear peptides and it could indeed be that the modification sites are mostly present in loop structures and disordered regions. Also only one particular peptide was studied. It is however a full protein that is ADP-ribosylated and it has a 3D structure that could affect the recruitment as well as accessibility. PARP14 is also able to automodify itself, even just the catalytic domain alone. How would these in vitro results be interpreted in the context of the current work? Therefore, the authors should discuss the limitations of the approach also to keep the proper balance in the conclusions. It would be of interest to look at the target protein list generated by a chemical genomics study the authors refer to whether such sequences appear in the putative PARP14 target proteins and whether these would indeed be a loop region based on experimental stuctures or e.g. AlphaFold predictions.”
Reply: We thank the reviewer for raising these concerns. We agree that a summary of the potential limitations of our study would be beneficial for balancing the concluding remarks. We have added a new paragraph that addresses each of the points they raise in the conclusion:
It is important to note that there are inherent limitations to the work we have described herein. Our study was designed to explore the minimal motifs required to guide proximal targeting by PARP14 which means we were only considering linear sequences and not considering the effects of folded proteins on targeting. It is certainly possible that the folded nature of various sites could influence their accessibility at the PARP14 active site. However, the recently proposed structure of PARP13 on AlphaFold has localized the P14p1 site to a disordered loop, which would be similar in structure to our analyzed peptides [31,32]. We also focused our study primarily on a single peptide. While we would argue that we have selected one of the more compelling PARP14 sites based on the previous literature, it will be important to explore how the proximal interactions we have identified are shared between future PARP14 sites and those found for other ARTD family members. Finally, P14c and the longer PARP14 variants undergo significant MARylation both in vitro and in vivo [24,30]. The presence of MARylation on these proteins could certainly alter target selection and it would be interesting to compare how proximal preferences are impacted by the loss of auto-modification sites.
We thank the reviewer for their feedback and suggestions. Their comments have led us to significantly revise our manuscript to clarify the scope of our work and to balance our conclusions to include potential limitations of the study. Their constructive criticism has helped us identify potential areas of data over-interpretation and allowed us to clarify our arguments. It has led to a stronger manuscript with enhanced controls in the form of P15c automodification that are now included in Figure S2b. Their input has helped us clarify the experimental results we present and should aid us in making our work more accessible to a larger audience.
Reviewer 2 Report
ADP-ribosylation (ADPr) is a reversible post-translational protein modification that regulates several cellular and biological processes such as DNA damage repair, cell proliferation and differentiation, metabolism, stress, and immunological responses. Diverse ADP-ribosyl transferases and hydrolases participate in the fine-tuning of ADPr systems to maintain cellular homeostasis. Diverse ADP-ribosyl transferases and hydrolases are involved in the fine-tuning of ADPr systems to maintain cellular homeostasis. Current manuscript identified probable sequential motif required for ADP-ribosylation using TLC-MALDI-TOF. Overall, it is very interesting approach and useful to understand the presence of particular amino acid play a critical role in regulating ADP-ribosyl transferases and hydrolases process. Author needs to rectify Figure numbering at line 248, it should be Figure 4 and in line 280 should be Figure 5.
Author Response
- “Author needs to rectify Figure numbering at line 248, it should be Figure 4 and in line 280 should be Figure 5.”
Reply: We thank the reviewer for pointing out this inconsistency. We have fixed the figure numbering as suggested.
We thank the reviewer for considering our manuscript. They have approved the manuscript for publication and have no further concerns.
Reviewer 3 Report
Javed and his colleagues conducted a study aimed at uncovering the specific sequences that are favored by PARP14 for target selection and hydrolysis. They adopted an in vitro approach, utilizing an 18-mer peptide previously marked by the PARP14 catalytic domain (P14c). This peptide served as a tool to identify the essential sequence elements required for recognition by P14c. Through a systematic analysis using the 18-mer peptide, the researchers pinpointed a minimal 5-mer sequence that is both necessary and sufficient for PARP14 targeting. Then they tested the sequence specificity through a systemic alanine mutation on each residue. This finding was validated using a closely related protein, PARP14 delta 553, which represents the full-length PARP14 protein.
Their investigation revealed that PARP14's targeting preference is characterized by a basic sequence, involving K/R-D/E motifs in proximity. Additionally, the study highlighted limitations in the prevailing non-enzymatic hydrolysis methods commonly employed to define ADP-ribose (ADPr) sites. It was found that factors such as pH and temperature impact the non-enzymatic removal of ADPr from substrates. The researchers then applied their approach to investigate enzymatic hydrolysis involving macroD1, macroD2, and TARG1 enzymes. Notably, the results demonstrated that macroD2 is primarily responsible for removing ADPr from the P14-modified peptide.
This manuscript exhibits a strong design and effective writing, incorporating appropriate controls. The results are clearly articulated, and the conclusions are succinct and well-aligned with their discoveries. Furthermore, this research introduces an additional means of delineating ADP-ribose (ADPr) site specificity for target selection, enhancing the toolkit available for such investigations.
Minor comments:
1. Please review Figure 3, which appears duplicated, and make corresponding adjustments to Figures 4 and 5 accordingly.
Author Response
- “Please review Figure 3, which appears duplicated, and make corresponding adjustments to Figures 4 and 5 accordingly.”
Reply: We thank the reviewer for pointing out this inconsistency. We have fixed the figure numbering as suggested. This was an oversight in our editing process.
We thank the reviewer for considering our manuscript. They have approved the manuscript for publication and have no further concerns.
Round 2
Reviewer 1 Report
The authors have addressed my comments and improved the manuscript.
Only a small point as they have maybe misunderstood the comment regarding the nomenclature. It is not necessary to use ARTD acronyms and actually PARP and TNKS names should be used and collectively they form the ARTD family. PARP is not to be used as an acronym (for poly-ADP-polymerase, which the enzymes are not). Please refer to the original publication: https://pubmed.ncbi.nlm.nih.gov/34323016/
From the scientific point of view the manuscript can be published.
Author Response
- “Only a small point as they have maybe misunderstood the comment regarding the nomenclature. It is not necessary to use ARTD acronyms and actually PARP and TNKS names should be used and collectively they form the ARTD family. PARP is not to be used as an acronym (for poly-ADP-polymerase, which the enzymes are not). Please refer to the original publication: https://pubmed.ncbi.nlm.nih.gov/34323016/.”
Reply: We appreciate the reviewer’s interest in aligning our manuscript with the agreed upon nomenclature in the field. This issue has been a challenge within the ADP-ribosyltransferase discipline and we are dedicated to making our manuscript clear not only to the specialists in our field, but to the broader audience who may never have heard of ARTDs or PARPs. To this effect, we have made sure to refer to the family of enzymes as ARTDs. We have kept the original PARP (or TNKS) numbering to refer to specific enzymes as agreed upon in the provided reference. We have also included a new sentence in the introduction that clarifies this issue to our audience, with an appropriate citation for the relevant reference.
We thank the reviewer for their helpful feedback throughout this process. Their comments have helped strengthen and improve our manuscript. We are pleased that our previous revisions “have addressed [their] comments and improved the manuscript,” and that “from the scientific point of view the manuscript can be published.” We hope that with the changes explicated above we have addressed their remaining concern regarding nomenclature, which will improve the legibility of our manuscript.